# Short-Term Variability of Non-Migrating Diurnal Tides in the Stratosphere from CMAM30, ERA-Interim, and FORMOSAT-3/COSMIC

**Subhajit Debnath** and **Uma Das** *

Indian Institute of Information Technology Kalyani, Kalyani 741235, India
* Correspondence: uma@iiitkalyani.ac.in

**Abstract:** The variability of non-migrating tides in the stratosphere is investigated using temperature data from Canadian Middle Atmosphere Model (CMAM30), ERA-interim reanalysis and Formosa Satellite-3 and Constellation Observing System for Meteorology, Ionosphere, and Climate (FORMOSAT-3/COSMIC) from 2006 to 2010 using a ±10-day window. CMAM30 and ERA results show that the amplitudes of non-migrating tides, DS0 and DW2, are negligible in the mid and high-latitude stratosphere, and the results from satellite datasets are significantly affected by aliasing in this region, in spite of using a smaller window size for analysis (±10 days). Significant short term variability ranging from 30 to 100 days is observed in DS0 and DW2 over the equatorial and tropical latitudes. These tides are seen as two prominent bands around the equator with DS0 maximising during boreal summers and DW2 maximising during boreal winters. These variabilities are compared with the variability in amplitude of the stationary planetary wave with wavenumber one (SPW1) in the high-latitude stratosphere using the continuous wavelet transform (CWT). It is found that during boreal winters, the variability of SPW1 at 10 hPa over 65° N is similar to that of DS0 and DW2 over the equator at 0.0007 hPa. This provides evidence that SPW1 from the high-altitude stratosphere moving upward and equator-ward could be interacting with the migrating diurnal tide and generating the non-migrating tides in the equatorial mesosphere and lower thermosphere (MLT). The variabilities, however, are not comparable during summers, with SPW1 being absent in the Northern Hemisphere. It is thus concluded that non-linear interactions could be a source of non-migrating tidal variability in the equatorial MLT region during boreal winters, but during summers, the tidal variabilities have other sources in the lower atmosphere. The anti-symmetric nature of the vertical global structures indicates that these tides could be the result of global atmospheric oscillations proposed by the classical tidal theory.

**Keywords:** non-migrating tides; wavelet transform; stratosphere; non-linear interactions

## 1. Introduction

Atmospheric tides are generated in temperature and wind fields due to the diabatic heating of the atmosphere [1–3] and wave–wave interactions [4–6]. Direct solar radiation absorption is the main cause of diabatic heating; for example, infrared radiation is absorbed by water vapour in the lower atmosphere, ultraviolet radiation is absorbed by the ozone layer in the stratosphere, and extreme ultraviolet radiation is absorbed by the thermosphere. These tides are classified into two types: (1) migrating tides, which propagate westward with same phase speed as the Sun—for example, DW1 (diurnal westward propagating tide with wavenumber one) and SW2 (semi-diurnal westward propagating tide with wavenumber two), and (2) non-migrating tides, which have different phase speeds whose apparent motion is either slower or faster than that of the Sun—for example, DS0 (diurnal stationary wave with zonal wavenumber zero) and DW2 (diurnal westward propagating wave with zonal wavenumber two). These tides are understood in terms of the solutions to the classical tidal theory based on linearised tidal equations [1–3].

Tidal variability in temperature and wind fields helps us to understand the dynamics of the atmosphere, including long-term and short-term variabilities in the atmosphere. The average global variability of migrating and non-migrating tides in the middle atmosphere has been established by investigations of various ground-based measurements [7–10] and satellite datasets [3,11–17], models [18,19], and reanalysis datasets [13,20]. Short-term tidal variability in the middle atmosphere is still not understood due to a lack of sufficient data from satellite and ground-based measurements. The diurnal oscillation has been observed with an amplitude of 1 to 1.5 m/s over Indonesia using radiosonde measurements of winds below 25 km [21]. Such studies based on a single location are mostly focused on the troposphere region. However, the migrating and non-migrating components cannot be resolved without simultaneous measurements over different longitudes. Even if such types of measurements are possible in a given latitude circle, all latitudes cannot be covered due to the land–sea distribution of the globe. On the other hand, satellites take global measurements (asynoptic) and are capable of resolving the migrating and non-migrating tides [22–24]. DW1 is found to exhibit a strong semi-annual variation with maxima during the equinox and minima during the solstice, as observed in Upper Atmosphere Research Satellite (UARS) measurements over low latitudes in the 50–120 km region [25]. Thermosphere, Ionosphere, Mesosphere Energetics and Dynamics (TIMED) satellite data have provided many insights into the tidal variability. The sounding of the Atmosphere using Broadband Emission Radiometry (SABER) instrument on-board the TIMED satellite can cover all 24 h local times over a given location in ∼60 days due to the yaw of the satellite [26,27]. Hence, ∼60 days of data are combined to extract tidal characteristics, thereby providing the average variability of the tidal field in that duration. Sakazaki et al. [13] showed that tidal amplitudes increase with altitude (up to 4 K in the lower mesosphere in SABER data) and also have a peak of 3.5 K at 1 hPa. Mukhtarov et al. [28] observed that the DW1 amplitudes maximised over the equator in the mesosphere region from 2002 to 2007, with amplitudes reaching up to 19 K at 90 km. In the stratosphere, the amplitudes increased towards mid-latitudes, reaching 4 to 5 K at the stratopause. The migrating diurnal tide shows distinctive features of the first symmetric propagating mode above 70 km height. They are related to the main maximum at the equator and secondary maximum near 35° latitude with a 180° phase shift. A trapped diurnal component is observed in the height range of 40 to 60 km [28]. The climatology of the diurnal tide was investigated using extended-CMAM data from 1979 to 2010 in the stratosphere and Mesosphere and Lower Thermosphere (MLT) region and compared with that obtained from SABER data [18]. During the Northern Hemisphere summer, DW1 has the largest amplitude below 65 km, but DE3 becomes the strongest in the MLT region. Semi-annual variation is observed at 95 km and annual variation is observed at 45 km in the SABER data. Spectral analysis of inter-annual variability shows a strong peak at 25 to 26 months in ext-CMAM and SABER data, which indicates the modulation of DW1 and DE3 tides by stratospheric Quasi Biennial-Oscillation (QBO) [18,29]. Although the average tidal variability is obtained in these studies, the background variations are found to alias into the diurnal migrating tide, DW1, thereby overestimating the amplitudes [13]. Aliasing can also take place between stationary planetary waves (SPW) and non-migrating tides; for example, the stationary planetary wave of wavenumber one (SPW1) can alias into DS0 and DW2 [14,30]. Other satellites such as Formosa Satellite-3/Constellation Observing System for Meteorology, Ionosphere and Climate (FORMOSAT-3/COSMIC) with a different sampling pattern are capable of addressing this aliasing problem and extracting short-term tidal variability in the Earth's atmosphere [14]. A shorter window size (±10 days) in analysing data helps to reduce the aliasing problem, but it still exists. Using numerical simulations, Das et al. [14] showed that non-stationary SPW1 variability causes more aliasing while a constant SPW1 variability results in zero aliasing.

The theory of tidal variability in the stratosphere is discussed in several papers [1–3], and non-migrating tidal characteristics were identified in the stratosphere from ground-based measurements [7,8,31], satellite observations of TIMED Doppler Interferometer (TIDI)

and SABER [28], UARS [32], the microwave limb sounder (MLS) [33], models [18], and reanalysis data [34]. Additionally, non-linear wave–wave interactions can also generate non-migrating tides [30]. SPW1 can interact non-linearly with DW1 and produce two child waves: DS0 and DW2 [12,19,35]. Observations show that the amplitudes of the non-migrating tides are very small in the high-latitude stratosphere and are believed to be produced due to such non-linear interactions between SPW1 and DW1 [24]. However, the amplitudes reported in this study using SABER data are unusually high. During the arctic winter of 2003–2004, DS0 and DW2 amplitudes were observed to range between 1 K and 3 K at 50 km over mid-latitudes [36]. DS0 and DW2 components in the wind field are found to vary from 5 to 15 m/s at 95 km in the Middle Atmosphere Circulation Model from Kyushu University (MACMKU), UARS, and Global Scale Wave Model (GSWM) data [30]. Recent numerical experiments show that non-linear interactions may not be an important source of the non-migrating tides in the high latitude stratosphere, as thought earlier [14], and are observed in SABER data due to aliasing [13,14,20]. Many studies also support the important role played by non-linear interactions in the generation of the non-migrating tides in the equatorial mesosphere. Over mid-latitudes, SPW1 is found to propagate upward and equator-ward and to generate DW2 via non-linear interaction with DW1 in the equatorial mesosphere [37]. Niu et al. [19] calculated the correlation between SPW1 and DW2 for the winter of 2009–2010 and found this to be 0.55. Their results also show significant correlations between SPW1 and DW2 amplitudes during 20 winters in 31 years from the period 1979–2010.

There is still a significant gap in understanding short-term variability in the middle atmosphere, particularly that of the non-migrating tides in the stratosphere, which cannot be addressed separately by satellite datasets due to the asynoptic nature of the measurements and associated aliasing problems. Thereby, models and reanalysis datasets have to be included for a better opportunity to investigate the tidal features in the atmosphere. With this motivation, non-migrating tides, DS0 and DW2, are investigated using data from CMAM30 and ERA-interim data in the stratosphere and compared with those obtained from COSMIC observations. The objective is to investigate their short-term variability and the importance of non-linear interactions in (1) the high-latitude stratosphere and (2) equatorial MLT region. Section 2 briefly describes the three datasets and the methodology, Section 3 describes the tidal characteristics determined, and Section 4 discusses the short term variability of the non-migrating diurnal tides including wavelet analysis and summarizes the results.

## 2. Data and Methodology

### 2.1. CMAM30 Model and ERA-Interim Reanalysis

The Canadian Middle Atmosphere Model (CMAM) is a global circulation model that provides a retrospective estimate of the chemical and dynamical evolution of the atmosphere from 1979 to 2010 [38–40]. There are two types of model outputs: 1. Regular CMAM30 (which extends up to 95 km), focusing on the troposphere–stratosphere–mesosphere region, and 2. Extended-CMAM (which extends up to 200 km), focusing on the MLT region [29,41,42]. CMAM30 provides model output data from 1000 hPa to $10^{-3}$ hPa and ext-CMAM provides data from 1000 hPa to $10^{-6}$ hPa every 6 h from 1979 to 2010 on a global grid of 96 longitudes and 48 latitudes. In the current study, our focus is on the stratosphere, and hence we used the regular CMAM30 model temperature data to examine the short-term variability of tides during the period from 2006 to 2010. A two-imensional fast Fourier transform (2D-FFT) is used to analyse temperature in the longitude ($\lambda$) and time ($t$) space to extract the different wave components using a window of $\pm 10$ days.

$$F(f,s) = \frac{1}{N} \sum_t \sum_\lambda f(t,\lambda) e^{-i2\pi(ft+s\lambda)} \tag{1}$$

The Fourier spectrum is a function of frequency ($f$) and wavenumber ($s$), where the frequency $f = 1$ gives information on the diurnal tides and frequency $f = 0$ on SPWs.

Positive wavenumbers represent westward propagating waves and negative wavenumbers represent eastward propagating waves.

The reanalysis data set, ERA-interim, is the largest global atmospheric reanalysis produced by the European Centre for Medium-Range Weather Forecasts (ECMWF) [43]. ERA-Interim is a global atmospheric reanalysis that is available from 1 January 1979 to 31 August 2019. For the current study, we investigate the temperature and wind data from 2006 to 2010 at 37 pressure levels at a spatial resolution of $3° \times 3°$. Similar to CMAM30 data analysis, ERA-Interim data are also analysed using the 2D-FFT method to extract the various wave components using a $\pm10$-day window [20].

The amplitudes of DS0, DW2, and SPW1 obtained from CMAM30 are analysed in detail using continuous wavelet transform (CWT) to find the short-term variability in the time–frequency space [44] in the stratosphere and MLT region. The Morlet function of degree 6 is used as the mother wavelet, which is basically a plane wave modulated by a Gaussian. The wavelet power spectra are investigated during the period of interest from 2006 to 2010.

### 2.2. FORMOSAT-3/COSMIC

FORMOSAT-3/COSMIC, a joint project of Taiwan and the United States, is a constellation of six small satellites [45] that probe the Earth's atmosphere using the GPS radio occultation (GPSRO) technique [22]. The mission was launched in 2006, and it provided observational data up to May 2020. The COSMIC "atmprf" dataset provides the dry temperature data in the stratosphere, and version 2.0 data are analysed in the current study. COSMIC data sampling is random and spatially non-uniform as it depends on the position of the COSMIC and the GPS satellites. COSMIC satellites make observations of the radio signals transmitted by the GPS satellites as they set or rise behind the Earth's horizon [22]. Hence, the observations are very random. Das et al. [14] have shown that a window size of $\pm10$ days is suitable for retrieving short-term tidal variability from such observations, and it also reduces the aliasing. For appropriate comparisons, CMAM30 and ERA-Interim data are also analysed using a window of $\pm10$ days.

The least square fitting method is used to investigate the short-term variability of the tides in the stratosphere during 2007 to 2010 using COSMIC data. All migrating and non-migrating diurnal tides are extracted along with the mean temperature variation and SPW components [14]. The following function is fit to the two-dimensional temperature data, $T$, a function of universal time, $t$, and longitude, $\lambda$, at a given latitude and altitude, to include (a) mean temperature ($T_0$) (b) tides and (c) SPWs.

$$T(t, \lambda) = T_0 + \sum_{i=1}^{3} \sum_{j=-4}^{4} T_{ij} \cos(2\pi f_i t + 2\pi s_j \lambda - \phi_{ij}) + \sum_{k=1}^{3} T_k \cos(2\pi s_k \lambda - \phi_k) \qquad (2)$$

where $T_{ij}$ and $\phi_{ij}$ are the amplitudes and phases of tides with frequency, $f_i$, taking values from 1 to 3 for diurnal, semi-diurnal, ter-diurnal tides and wavenumber, $s_j$, ranging from $-4$ to $+4$, where negative (positive) wavenumbers denote eastward (westward) propagating tides. $T_k$ and $\phi_k$ are the amplitudes and phases of the SPW with wavenumbers, $s_k$, ranging from 1 to 3. As already mentioned, a window size of $\pm10$ days is used to extract the short-term wave variability, i.e., data over $\pm10$ days are combined to obtain the wave features and are attributed to the central day [14]. The process is repeated at each altitude from 15 to 50 km at all latitudes from 70° S to 70° N. COSMIC data have been validated, and it has been found that they are reliable up to approximately 45 km [46].

## 3. Tidal Characteristics of DS0 and DW2

### 3.1. Altitude–Time Variations

Amplitudes and phases of the non-migrating diurnal tide, DS0, extracted from COSMIC, CMAM30, and ERA temperature data over the equator during 2006 to 2010 are shown in Figure 1. The amplitudes observed in COSMIC are small, varying from 0.6 to 0.9 K,

and observed intermittently from 35 to 40 km. Above 40 km, the amplitudes are almost negligible. In CMAM30 and ERA, on the other hand, significant amplitudes greater than 0.6 K are seen above 5–10 hPa. The year-to-year variations are consistent within these datasets in both amplitude and phase. There is a seasonal variation with maxima during the month of August and minima during the month of October. Amplitudes of 1 K are observed during August 2006, 2008, and 2009 in both CMAM30 and ERA at altitudes above 3 hPa. The amplitudes are comparatively smaller in 2007. These features are absent in COSMIC. However, the phase variations are highly consistent among all three datasets above 30 km/10 hPa.

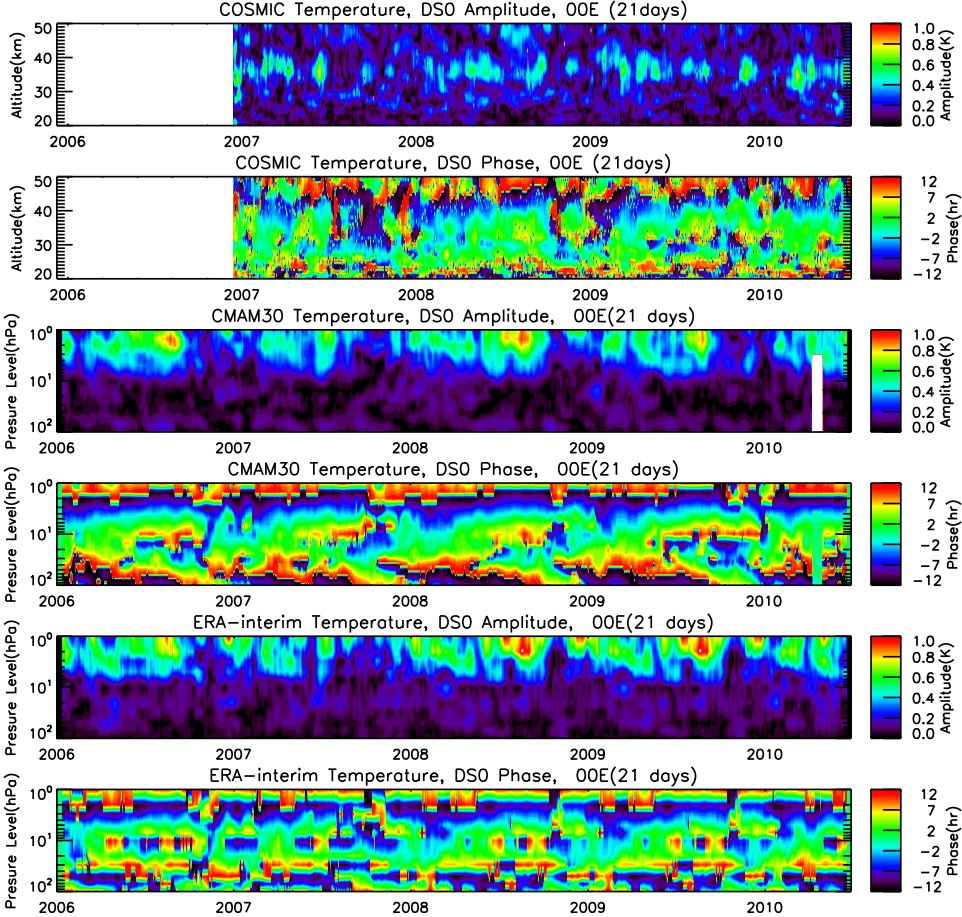

**Figure 1.** Amplitudes and phases of DS0 tide over equator from COSMIC observations, CMAM30, and ERA-interim reanalysis datasets using ±10 days window.

Figure 2 shows the variability in DS0 amplitudes and phases over 65° N similar to Figure 1. Significant variability is observed in COSMIC during the boreal winter. Tidal amplitudes of 1 to 2 K are observed from 25 to 40 km in small bands varying periodically over scales of less than a month. In CMAM30 and ERA, short-term variations, albeit with small amplitudes, are also seen during the boreal winters. During the winters of 2008–2009 and 2009–2010, these amplitudes are slightly larger. Earlier studies have shown that non-stationary, i.e., changing SPW1 amplitudes can alias into DS0 and DW2 components [13,14,30]. Das [37] has shown that SPW1 varies with similar periodicities from 30 to 120 days. The DS0 amplitudes observed in Figure 2 in COSMIC, thereby, appear to arise due to aliasing. The phases in CMAM30 and ERA, although highly consistent and similar, do not provide any useful information as the amplitudes are very small.

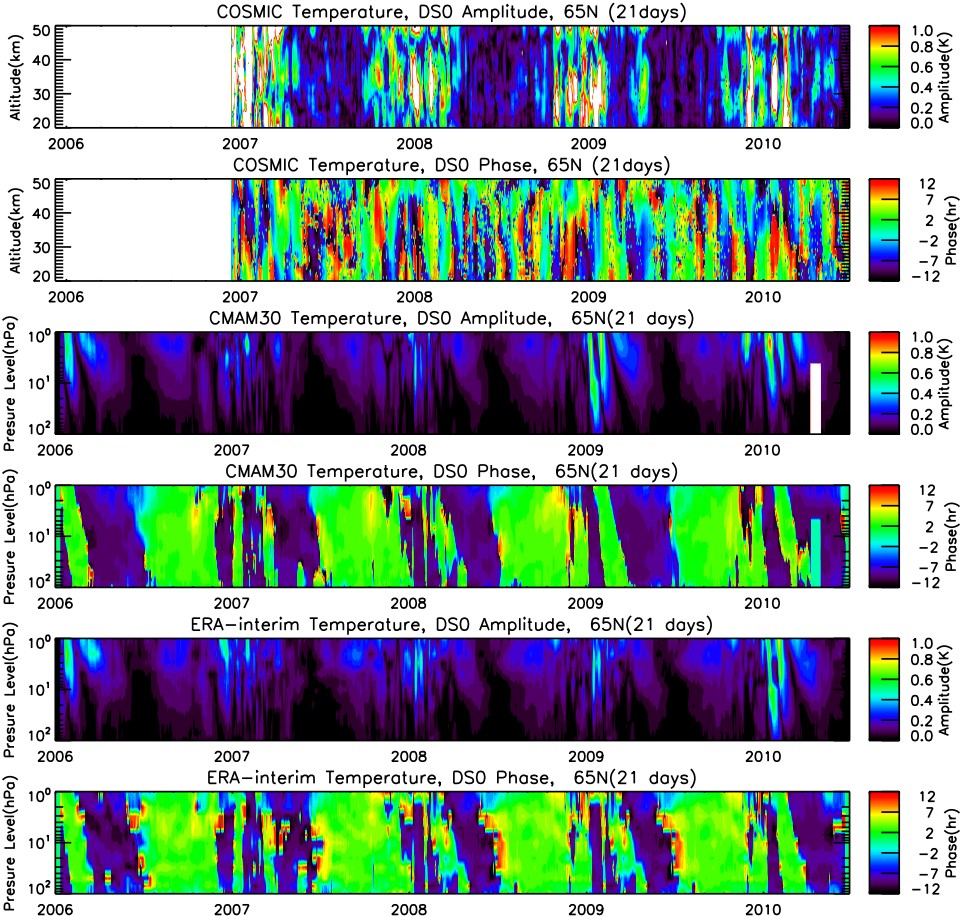

**Figure 2.** Amplitudes and phases of DS0 tide over 65° N from COSMIC observations, CMAM30, and ERA-interim reanalysis datasets using ±10 days window.

Amplitudes and phases of the DW2 tide extracted from COSMIC, CMAM30 and ERA temperature data over the equator are shown in Figure 3. Similar to DS0, the variability of DW2 in COSMIC temperatures is intermittent and observed between 30 and 40 km. In CMAM30 and ERA, regular variations with a winter maximum are seen above 5 hPa. CMAM30 amplitudes at 2 hPa are 1 K, and ERA amplitudes are larger and reach values of 2 K. Further, in ERA data, DW2 shows occasional peaks during summers indicating a semi-annual variation superimposed on an annual variation. DW2 phase variations of CMAM30 and ERA are similar and comparable with that of COSMIC.

Figure 4 shows the variability of DW2 amplitudes and phases over 65° N. Similar to DS0, the variations in DW2 amplitudes also appear to be affected significantly by aliasing. Amplitudes of 0.8 to 1 K are observed in COSMIC temperatures from 25 to 40 km during boreal winters as striations vary periodically over scales of less than a month and are probably due to aliasing, as seen in the case of DS0 [13,14,30]. DW2 amplitudes are almost absent over 65° N in CMAM30 and ERA temperature data. Phases of DW2 over 65° N do not provide any information in any of the datasets.

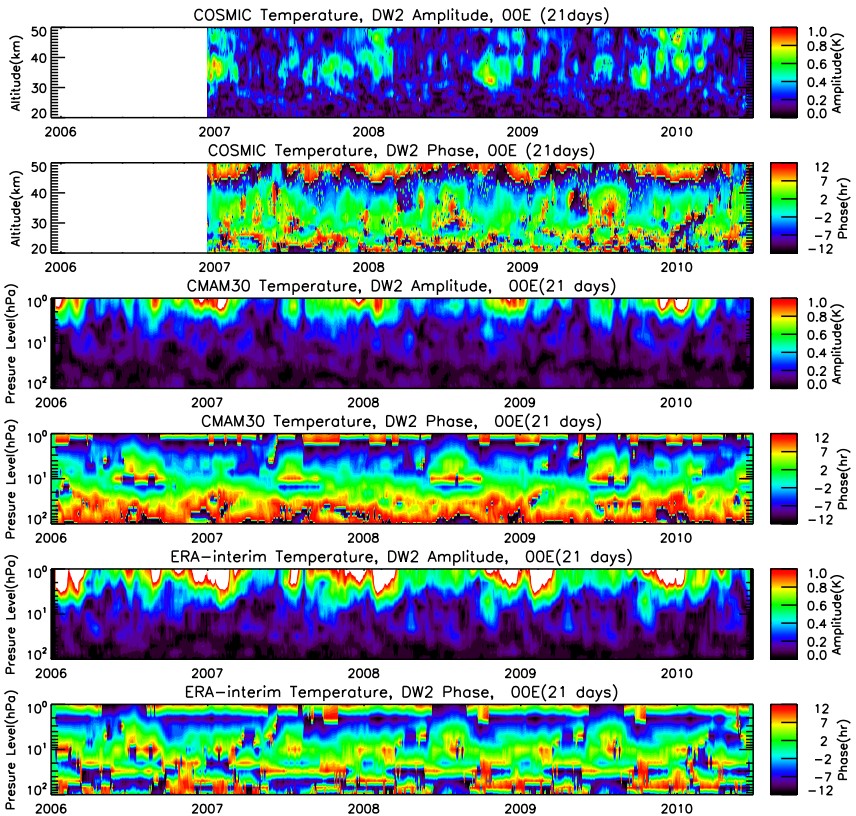

**Figure 3.** Amplitudes and phases of DW2 tide over equator from COSMIC observations, CMAM30, and ERA-interim reanalysis datasets using ±10-day window.

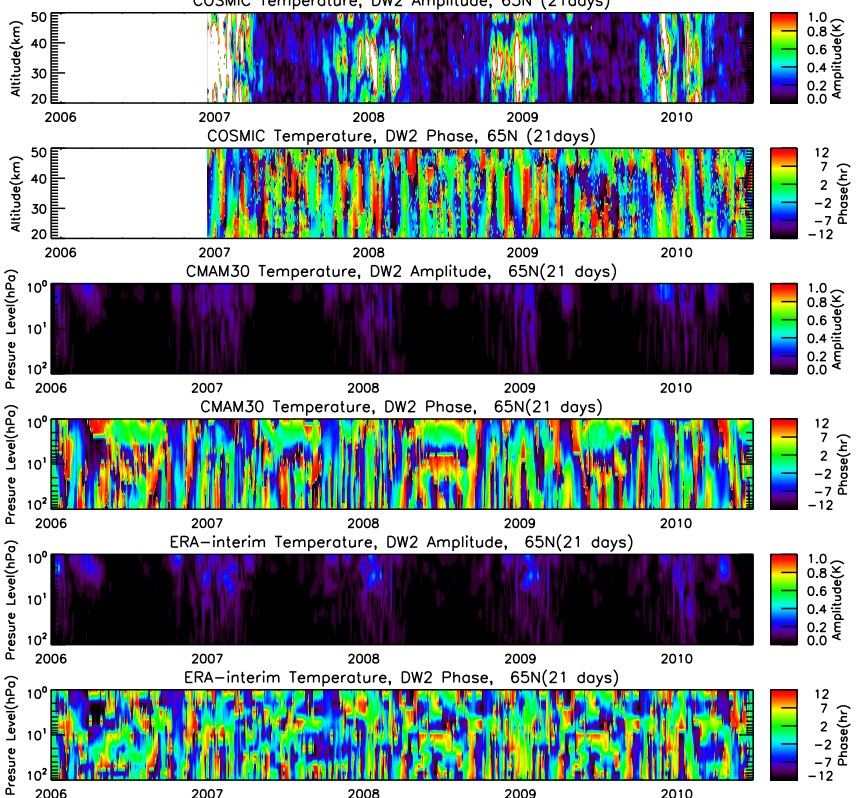

**Figure 4.** Amplitudes and phases of DW2 tide over 65° N from COSMIC observations, CMAM30, and ERA-interim reanalysis datasets using ±10-day window.

### 3.2. Altitude–Latitude Variations

Monthly means of amplitudes and phases of the DS0 tide as a function of altitude and latitude from COSMIC, CMAM30, and ERA temperature data are shown in Figure 5, in the three columns, respectively. The first and second rows correspond to amplitudes and phases, respectively, during August 2008, representing boreal summer, and the third and fourth rows correspond to those during February 2009, representing boreal winter. The DS0 tide shows an amplitude of 1 K at 35 km over 5° N and 15° S during the summer of 2008 from COSMIC observations. Beyond 40 km, the amplitudes are negligible. This feature is not reliable due to the lower quality of COSMIC data above 40–45 km. The corresponding phase plot from COSMIC data shows that the two maxima are out of phase and anti-symmetric with respect to the equator. The vertical wavelength is approximately 20 km. In CMAM30 temperature data, DS0 shows an amplitude of 1 K at 1 hPa over 5° N and an amplitude of 0.6 K over 15° S during the summer of 2008. Amplitudes extend up from 10 hPa and above over the sub-tropical latitudes. DS0 shows secondary peaks with an amplitude of 0.6 K over ±35° latitudes in both hemispheres, from altitudes of 3 hPa and above. Phases of DS0 indicate that the two maxima over the tropical latitudes are out of phase and anti-symmetric with respect to the equator with a vertical wavelength of 20 km, and those of the secondary peaks are in phase. In ERA temperature data, The DS0 tide shows slightly larger amplitudes of 1.5 K at 1 hPa over 5° N and 15° S during the summer of 2008. DS0 also shows second peaks with an amplitude of 0.6 to 1 K over 35° N and 40° S during the summer of 2008. These amplitudes are significant above 5 hPa. The phase of DS0 indicates that the two maxima over the tropical latitudes are out of phase and anti-symmetric with respect to the equator within a range of ±30° latitudes and in phase over ±40° latitudes. Vertical wavelengths are also similar to that of CMAM30. During winter, the amplitude of DS0 over these sub-tropical latitudes becomes negligible, and large amplitudes are observed in COSMIC in the high latitudes of the Northern Hemisphere. The vertical wavelength of this component is much larger. In CMAM30, the DS0 tide shows a peak with an amplitude of 0.6 K over the equator from CMAM30 data, and similar amplitudes are observed in the high latitudes of the Northern Hemisphere. The phase of DS0 is symmetric in nature with respect to the equator within the range of ±40°, and the vertical wavelength over the equator is approximately ∼30 km. In ERA temperature data, winter DS0 variability is slightly different. It shows a broad peak with an amplitude of 0.6 K over ±20° latitudes. The phase plot is similar to CMAM30 and symmetric with respect to the equator. The vertical wavelength is approximately 30 km in CMAM30 and ERA-Interim during winter.

The vertical and global structures of the DW2 tide showing the altitude–latitude variability during the summer of 2008 and winter of 2008/09 are shown in Figure 6. DW2 amplitudes are relatively smaller with an amplitude of 0.6 to 0.8 K at 35 km over 25° S in the Southern Hemisphere during the summer of 2008. A much smaller and negligible peak is observed over 5° N in the Northern Hemisphere. These two maxima are also anti-symmetric with respect to the equator. In CMAM30 temperature data, the DW2 amplitude is relatively smaller with values reaching 0.6 K over the equator during summer of 2008. In ERA-interim temperature data, DW2 shows a peak with an amplitude of 0.8 K over 5° S at 1 hPa. The phases are anti-symmetric with respect to the equator in both datasets, and the vertical wavelengths are approximately 25 km. During winter, DW2 shows negligible amplitudes over the tropical latitudes and high amplitudes in the high latitudes of the Northern Hemisphere. The amplitude is small, reaching values of 0.5 K at 5° S at 1 hPa in CMAM30, and much larger in ERA, reaching values of ∼2 K at 1 hPa over the equator and an amplitude of 1 K at 10 hPa over 10° N during winter. There appear to be two bands emanating and going upward around the equator, with a stronger Southern Hemisphere band in the both CMAM30 and ERA. The phase of DW2 indicates that two bands over the tropical latitudes are out of phase and anti-symmetric in nature within a range of ±25° latitudes, and the vertical wavelength is approximately ∼20 km. This clearly indicates that CMAM30 and ERA results for mean DS0 and DW2 variability are highly consistent with

each other in the stratosphere. COSMIC compares well in terms of vertical wavelengths over the tropical altitudes, but not in terms of amplitudes. Above 40–45 km, the data quality poses problems, and in the high latitude regions, aliasing poses problems.

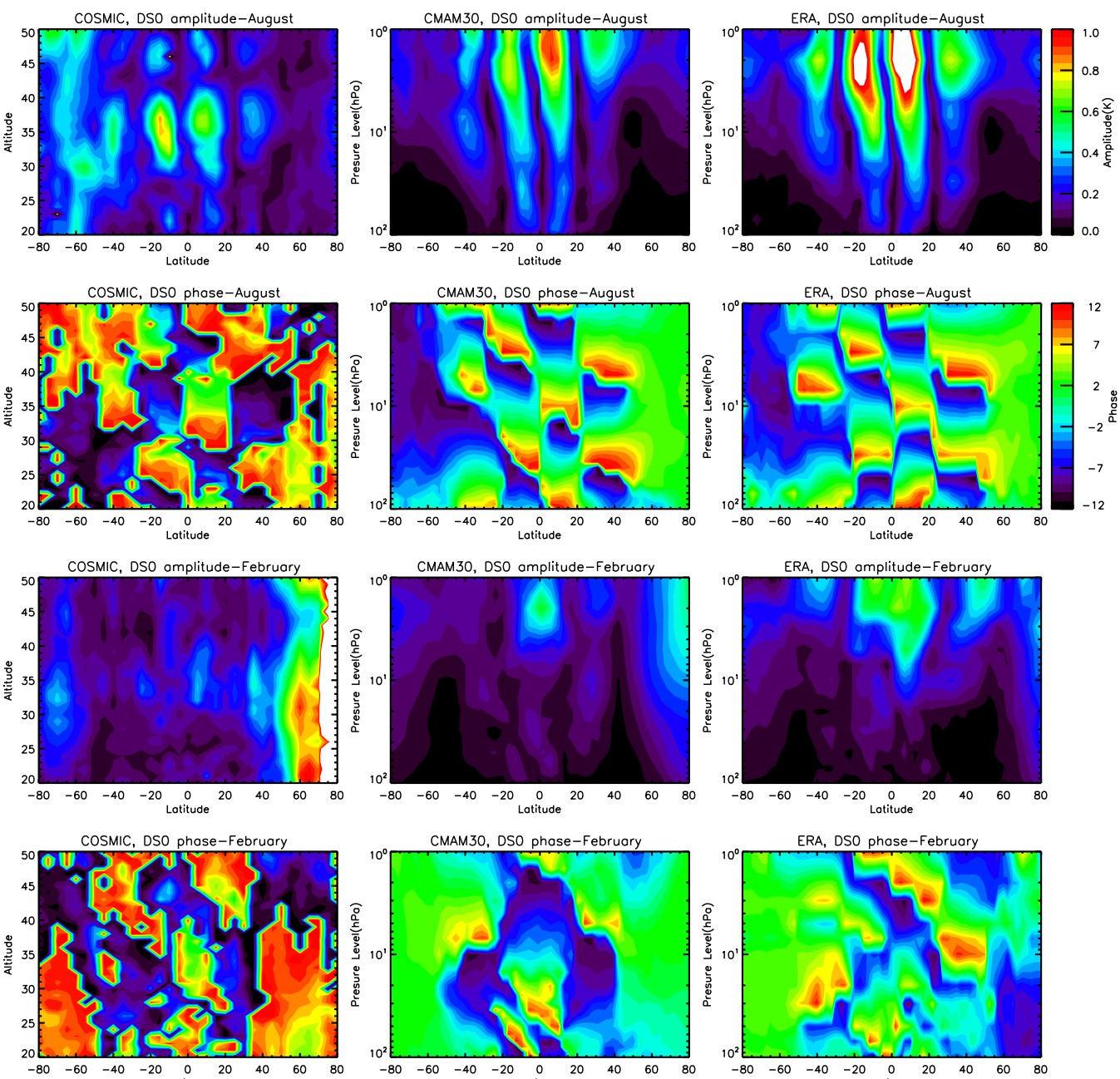

**Figure 5.** Altitude versus latitude cross-section of mean amplitude and phase of DS0 tide during August 2008 and February 2009 from COSMIC, CMAM30, and ERA-interim data.

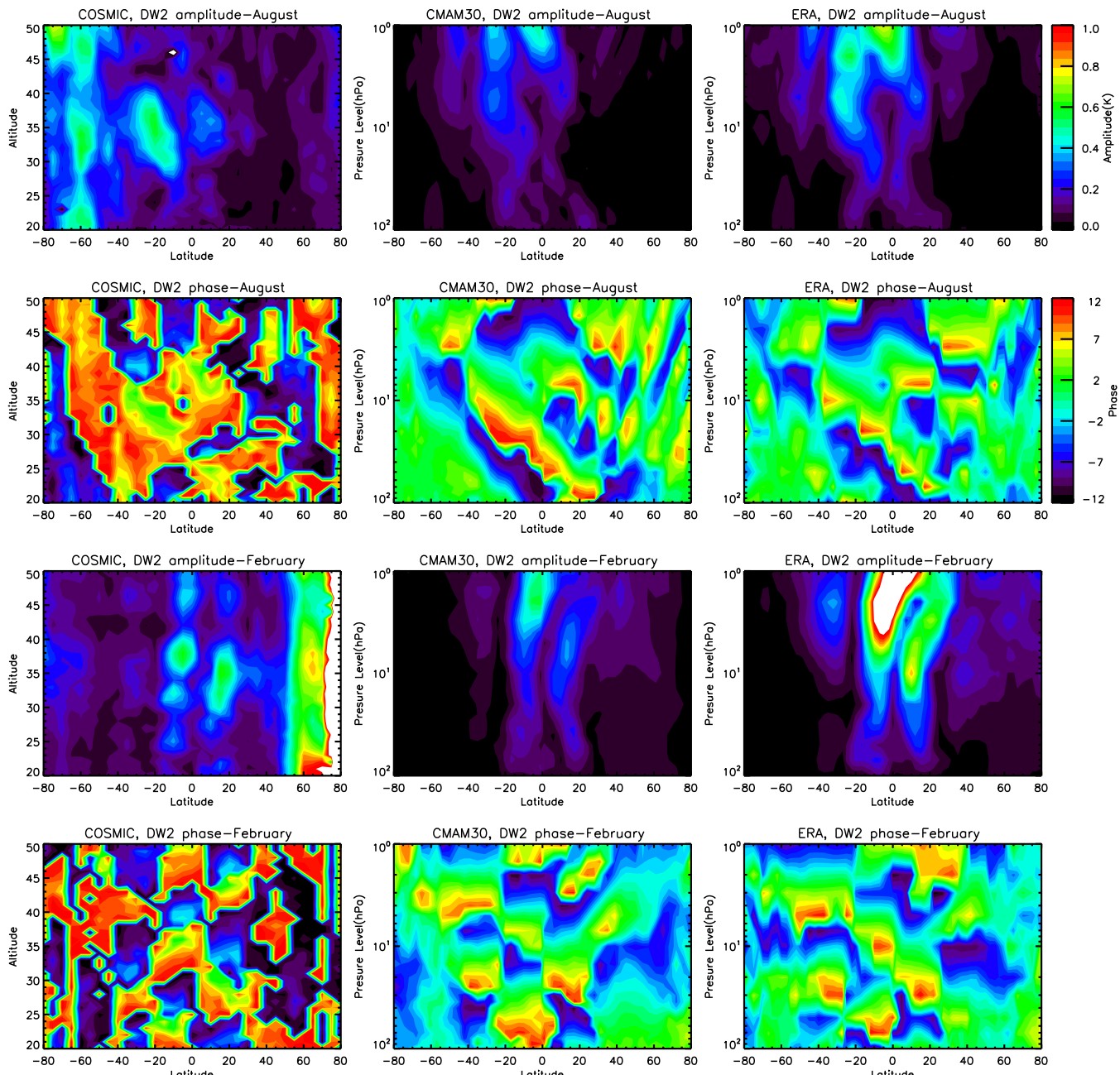

**Figure 6.** Altitude versus latitude cross-section of mean amplitude and phase of DW2 tide during August 2008 and February 2009 from COSMIC, CMAM30, and ERA-interim data.

### 3.3. Latitude-Time Variations

Tidal amplitudes of DS0 and DW2 from COSMIC at 30 km and from CMAM30 and ERA temperature data at 10 hPa are compared to understand the horizontal structure of their variability. The amplitudes of DS0 are shown in the top three rows of Figure 7 and those of DW2 are shown in the bottom three rows. In COSMIC data, a maximum amplitude of 0.4 to 0.6 K at ±10° latitudes is observed during summers. Larger amplitudes reaching values of 2 K are observed during winters. In CMAM30 and ERA data, the tropical band-like structures with an amplitude of 0.5 to 0.6 K at ±10° are also observed for DS0. The amplitudes of ERA are slightly larger in comparison, reaching 0.8 K. However, there are no significant amplitudes seen over winter at high latitudes, as observed in COSMIC. Additionally, in the Southern Hemisphere, a third band-like structure is seen in CMAM30 and ERA, around 35° S during summer, which gradually extends into the

high-latitude region by the September/October months. No such feature is seen in the Northern Hemisphere. It is also not seen in COSMIC observations in both hemispheres.

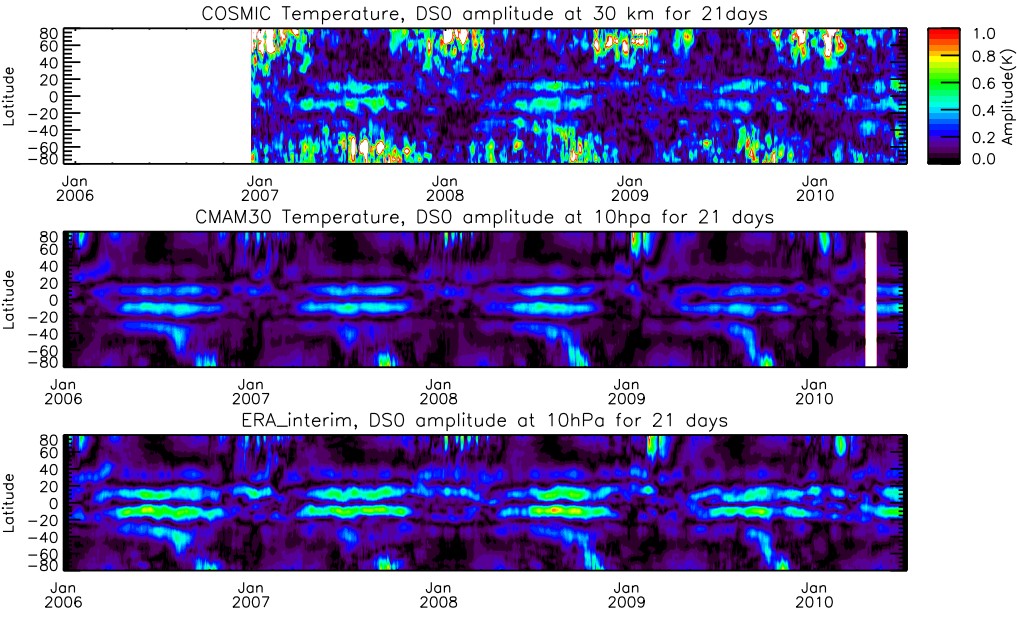

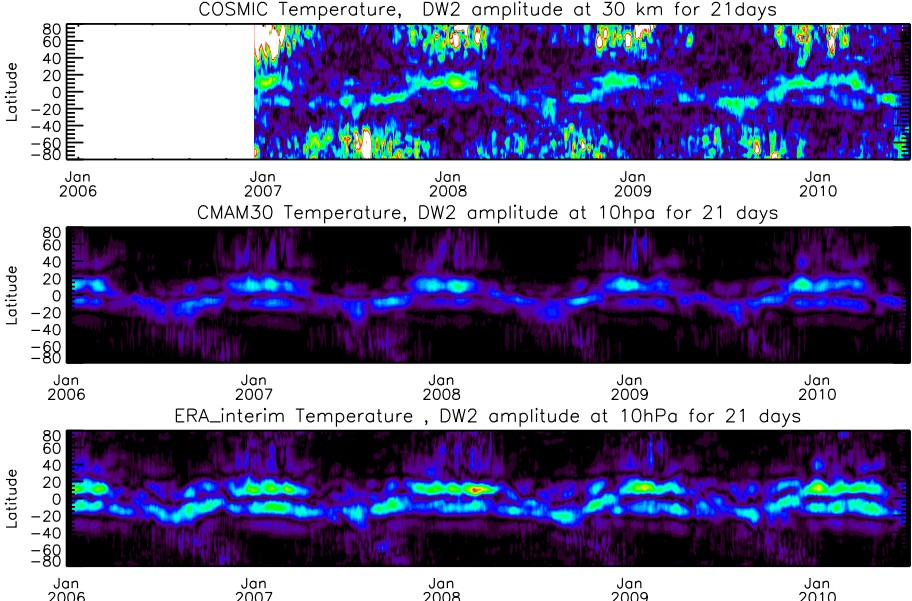

**Figure 7.** Latitude versus time variation of amplitudes of non-migrating tides, DS0 (in **top** three panels), and DW2 (in **bottom** three panels) at 30 km from COSMIC observations, and at 10 hPa from CMAM30 and ERA-interim data obtained using a 21-day window.

DW2 tidal amplitudes also show a similar structure with two tropical bands over the ±10° latitudes. However, one of the bands is stronger in a given period. During boreal winter, the band in the Northern Hemisphere is stronger in COSMIC and CMAM30, while both bands are of almost similar amplitudes in ERA-Interim. During boreal summers, DW2 amplitudes are much smaller. Similar to DS0, the DW2 amplitudes are larger, reaching values of 2 K over winter high-latitude regions in COSMIC, but are negligible in CMAM30 and ERA.

## 4. Discussion

It is very difficult to extract all tidal features from satellite measurements with accuracy and no aliasing due to the asynoptic nature of measurements, particularly over mid and high-latitudes. Hence, simultaneous investigations of models is essential to correctly determine the tidal characteristics. The current study aimed to extract the short-term tidal variability along with an attempt to avoid aliasing by using multiple datasets from satellite, model, and reanalysis.

The non-migrating tidal variability in the stratosphere from CMAM30 and ERA seen in the current study matches very well both in terms of amplitude and phase. Variability in COSMIC, however, differs, particularly in the high latitudes. These regions are significantly affected by aliasing problems in the analysis of satellite datasets [13,14,20,30]. Das et al. [14] showed that using a shorter time window length using COSMIC data, the aliasing is significantly reduced, but still exists. The non-migrating tides, DS0 and DW2, are particularly affected by the non-stationarity in the amplitude of SPW1. Das [37] showed that SPW1 amplitudes in SABER and COSMIC temperatures show a variability of 30 to 120 days in the high-latitude stratosphere. The smaller period variabilities were seen in COSMIC using a window size of 21 days and thus clearly indicate that SPW1 is non-stationary. Thereby, it can impact the accurate deduction of DS0 and DW2 variabilities. It is thereby concluded that there is high probability that the high DS0 and DW2 amplitudes observed in the high-latitude stratosphere in COSMIC could be due to aliasing. This argument is further strengthened by the results from CMAM30 and ERA that do not suffer from aliasing problems and do not show any significant DS0 and DW2 amplitudes in the high-latitude stratosphere. Many studies showed that the non-migrating tides, DS0 and DW2, can be generated in the middle atmosphere via non-linear interactions between DW1 and SPW1 [24,30]. However, the amplitudes of DS0 and DW2 deduced from SABER/TIMED are highly overestimated [24], largely owing to aliasing issues [14,20]. These latter studies established significant doubt on the process of generation of DS0 and DW2 via non-linear interactions, as such large amplitudes or variabilities are not observed in the current study using CMAM30 or ERA data.

In another study by Lieberman [35] using SABER/TIMED data, DW2 variability in the MLT region over the equator was found to be very similar to that of SPW1 over the high-latitude stratosphere. It was inferred that the high-latitude SPW1 in the stratosphere propagates upward and equator-ward, where it interacts non-linearly with the DW1 tide and generates DW2. DS0 was not observed. In the current study, DS0 and DW2 variabilities are investigated using CMAM30 data to identify similarities with that of SPW1 (ERA and COSMIC data are not available in the MLT region and hence not investigated further). Figure 8 shows the latitude–time variabilities of amplitudes of DS0, DW2, and SPW1 at 0.0007 hPa (in the top three panels) and 2 hPa (in the bottom three panels), respectively, from CMAM30 data using a 21-day window. DS0 and DW2 amplitudes at 2 hPa are slightly larger and shifted northward in comparison to those seen at 10 hPa in Figure 7. The central band of DS0 over the equator dominates, with larger values during boreal summer, and the structure appears slightly distorted in winter. In DW2, both bands become prominent, with larger values during boreal winter, and the structure appears to be distorted during boreal summers. Simultaneously, SPW1 at 2 hPa dominates the winter high latitudes with amplitudes ranging from 10 to 20 K and is stronger in the Northern Hemisphere. It is very clearly seen that at the stratospheric altitudes over the low-latitude region, DS0 and DW2 amplitudes are significant and there is no one-to-one correlation with SPW1 in the higher latitudes. This shows that the non-migrating tides could have been produced by the basic physical processes considered by the classical tidal theory. At a higher altitude of 0.0007 hPa (Figure 7), SPW1 amplitudes reduce and show peaks with amplitudes of 4–5 K over the winter high latitudes and over the equator the amplitudes are in the range of 2–3 K. Simultaneously, DS0 and DW2 tides show maxima with amplitudes of 4–5 K during boreal summer and winter, respectively. Additionally, the variability of DS0 is weaker and distorted during winters, while that of DW2 during summers is almost negligible. For both

tides, the band over the equator is strongest. All amplitudes show significant short-term variabilities of the order of a month.

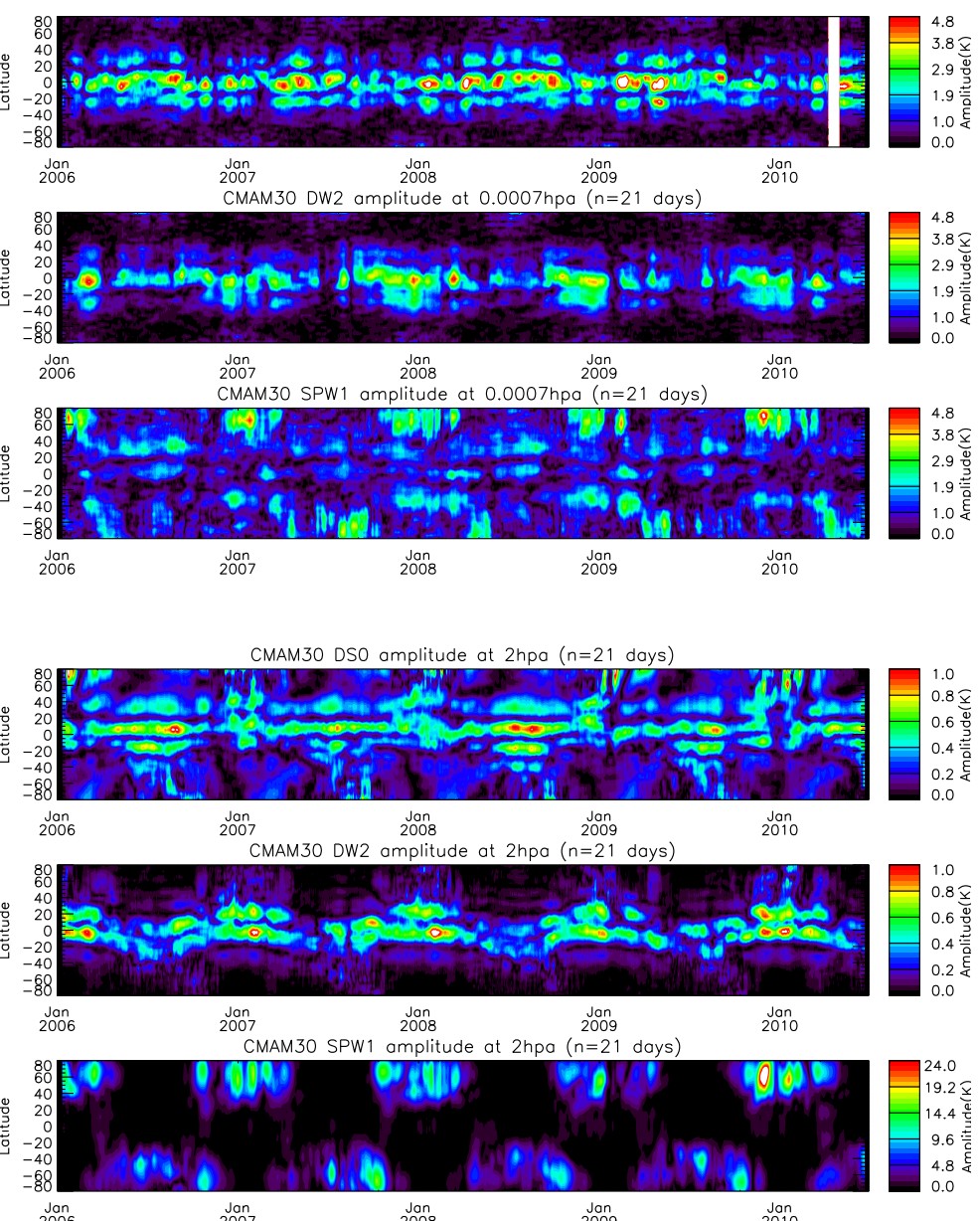

**Figure 8.** Latitude versus time variation of amplitudes of DS0, DW2, and SPW1 tide at 0.0007 hPa (∼95 km, in the **top** three panels) and 2 hPa (∼40 km, in the **bottom** three panels) from CMAM30 temperature data.

We perform a wavelet analysis using CWT with a Morlet wavelet of degree 6, i.e., $\omega_0 = 6$ [44], to investigate these variabilities among the different waves. Figure 9 shows the wavelet power spectrum of the average amplitude of DW2 over ±20° at (a) 0.0007 hPa, (b) 1 hPa, and (c) 10 hPa, along with (d) that of SPW1 amplitude over 65° at 10 hPa from CMAM30 temperature data. The black contours show the wavelet power at a confidence level of 99% with respect to a white noise background [44]. In all panels, the periods at 50 and 100 days are highlighted using horizontal dashed lines to aid the eye. The SPW1 amplitude in the high-latitude stratosphere shows significant variabilities during boreal winters at periods of 70 days in 2006–2007, 100 days in 2007–2008, 60 days in 2008–2009, and 70 days in 2009–2010. Apart from these, there is a prominent annual variation, which is expected due to the high amplitudes observed every winter. Amplitudes of DW2 over the

equatorial region, particularly in the higher altitudes, show similar periods during boreal winters and at other times, and there is no correlation with SPW1 variability. At 10 hPa, a weak semi-annual variation and a strong annual variation are observed. Our interest is in periods less than 100 days. A small island of statistically significant power is seen in January 2007 at 30 days. Periods of 70–80 days are observed from mid-2007 to mid-2008, and those from 50 to 100 days are observed during the winter of 2008–2009. More small islands are seen at 50 days in September–October of 2008 and 40 days in November–December 2009. During the winter of 2009–2010, 100-day variability is also seen. At 1 hPa, a 50-day variation is seen from mid-2006 to almost the end of 2007, followed by 100-day variability from September–October of 2008 to September–October of 2009, and a 50-day period is once again indicated during the winter of 2009–2010. Further up at 0.0007 hPa ($\sim$95 km), the situation is more complex, with dominant periods increasing from 50 days to 100 days from mid-2006 to mid-2008. From mid-2008 onwards, the semi-annual and $\sim$70 days become prominent. Early 2010 also sees a $\sim$70-day period. Comparison between high-latitude stratospheric SPW1 amplitudes and high-latitude equatorial DW2 amplitudes shows that similar periods exist in both these waves, mostly during boreal winters, and there is no similarity during other epochs.

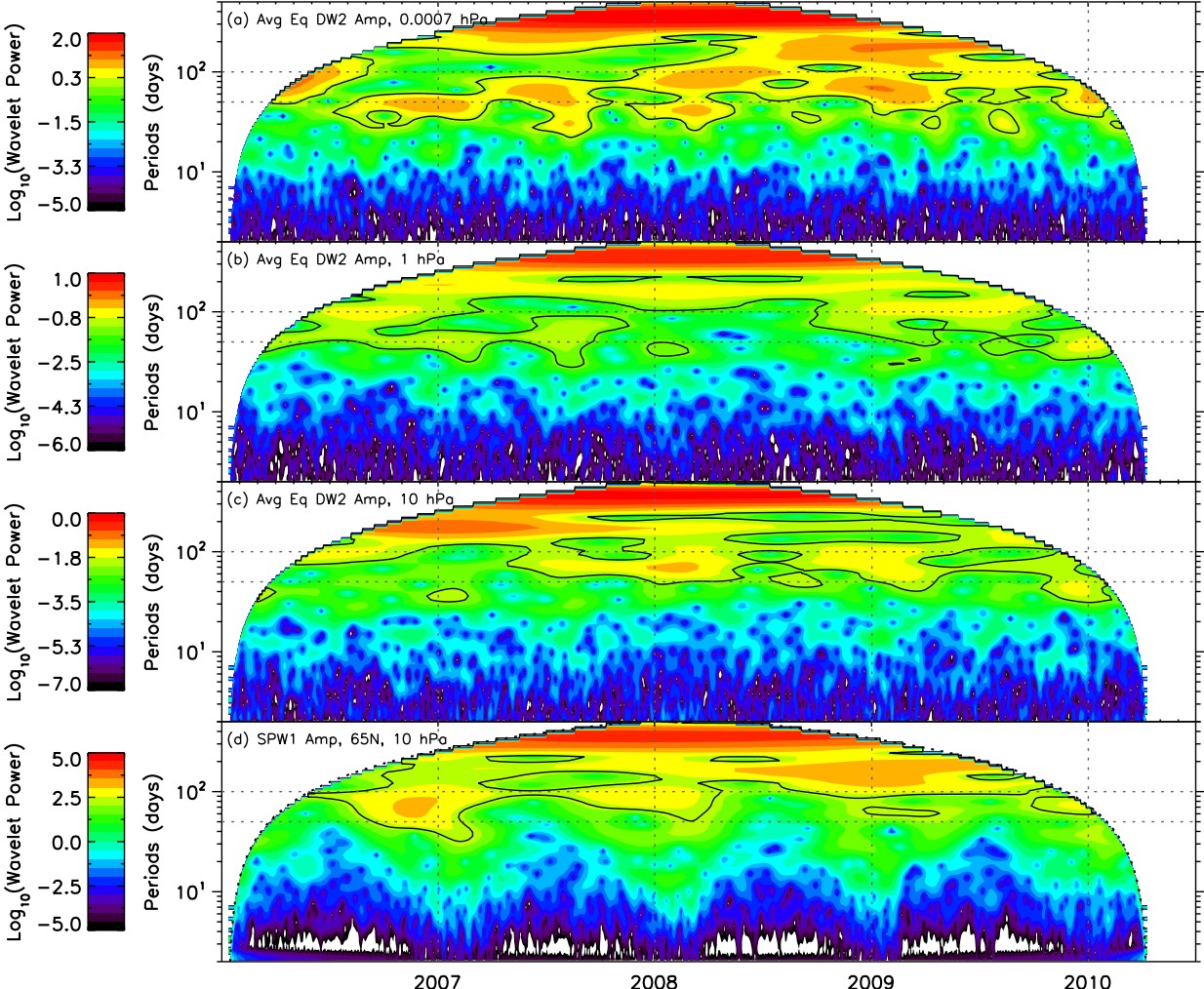

**Figure 9.** Wavelet power spectrum of average amplitude of DW2 over $\pm 20°$ at (**a**) 0.0007 hPa, (**b**) 1 hPa and (**c**) 10 hPa, (**d**) wavelet power spectrum of SPW1 amplitude over 65° N at 10 hPa extracted from CMAM30 temperature data.

Similar analysis is also performed for the amplitude of DS0. Figure 10 shows the wavelet power spectrum of the average amplitude of DS0 over ±20° at (a) 0.0007 hPa, (b) 1 hPa, and (c) 10 hPa, along with (d) that of the SPW1 amplitude over 65° N at 10 hPa from CMAM30 temperature data (similar to last panel in Figure 9). The black contours show the wavelet power at a confidence level of 99% with respect to a white noise background [44]. In this figure also, in all panels, the periods at 50 and 100 days are highlighted using horizontal dashed lines to aid the eye. Unlike in DW2, the short-term variability in DS0 in the stratosphere (10 hPa) is weak. One hundred-day variability is seen from September 2008 to September 2009. The annual oscillation is very prominent. At 1 hPa, 50-day variability is seen from mid-2006 to almost end of 2007, similar to DW2. Higher up, at 0.0007 hPa, significant variability is seen at periods from 40 to 100 days at the end of 2006, summer of 2007, beginning of 2008, and winter of 2008–2009 as well. This also is similar in many ways to the variability seen in DW2.

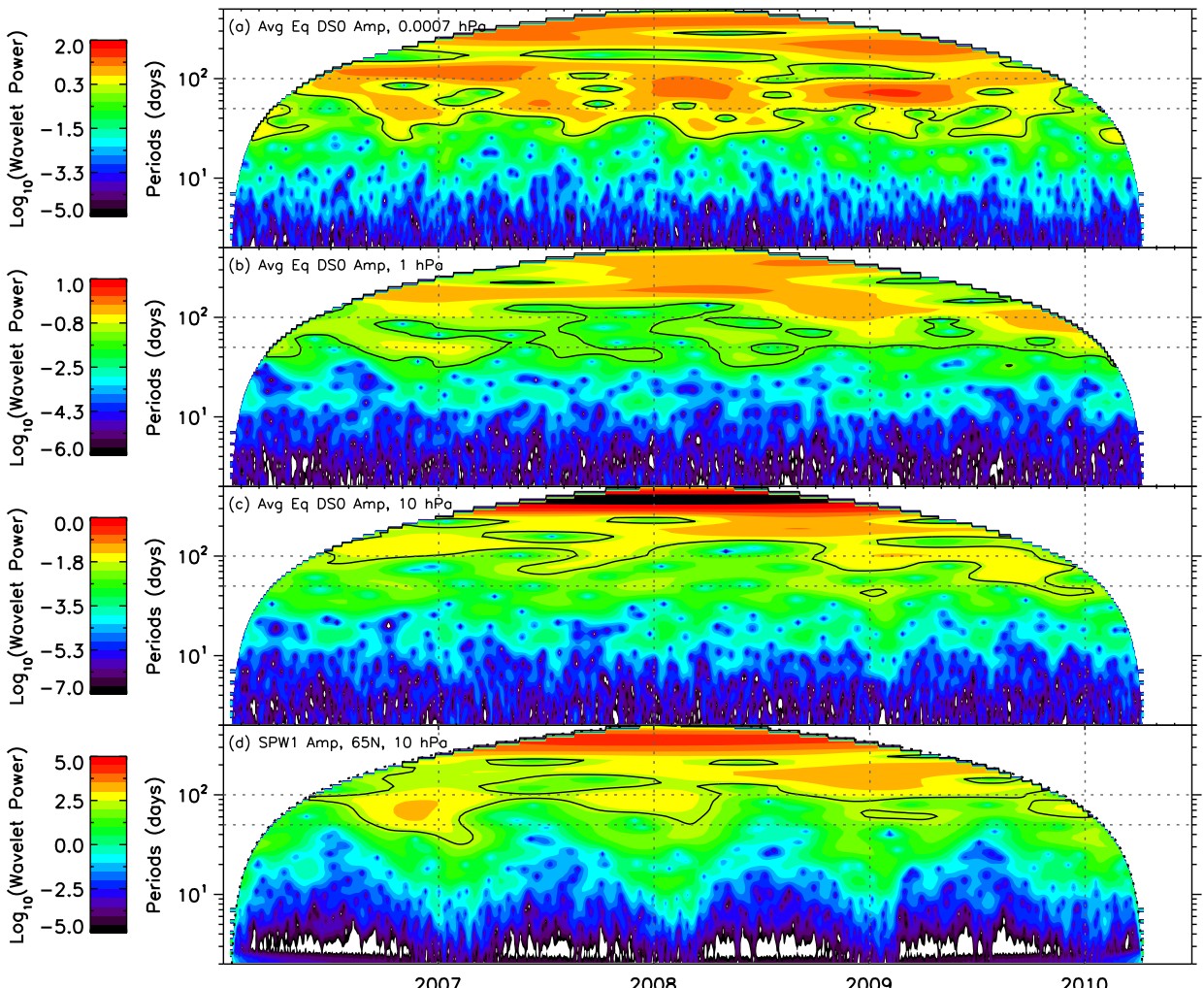

**Figure 10.** Wavelet power spectrum of average amplitude of DS0 over ±20° at (**a**) 0.0007 hPa, (**b**) 1 hPa and (**c**) 10 hPa, (**d**) wavelet power spectrum of SPW1 amplitude over 65° N at 10 hPa extracted from CMAM30 temperature data.

Detailed investigation of the Figures 9 and 10 shows that variability in SPW1 amplitude in the high-latitude stratosphere is occasionally similar to that in DS0 and DW2 amplitudes in the equatorial high-altitude, i.e, the MLT region—~50-day period in the winter of 2006–2007, ~100-day period in the beginning of 2008, and 70-day period in the beginning of 2009. At other occasions, such as during the summer of 2007, and at periods less

than 50 days, variabilities seen in DS0 ad DW2 are not observed in SPW1 amplitude (the variabilities in SPW1 amplitude over the Southern Hemisphere at 65° S are also verified (not shown here), and no similarities are seen). We thereby conclude that this is supporting evidence that part of the variability of DS0 and DW2 in the equatorial MLT region during winter could be attributed to non-linear interactions, while variability over significant durations otherwise, particularly during summers, remains unexplained.

A simple correlation analysis is performed between DS0 and DW2 amplitudes over the equator from 2006 to 2010 and that of SPW1 in the high-latitude stratosphere to investigate this further. In Figure 11, panel (a) shows the correlation between amplitudes of high-latitude stratospheric SPW1 with equatorial DS0, panel (b) shows the correlation between equatorial DS0 at 10 hPa with equatorial DS0 at all altitudes, panel (c) shows the correlation between amplitudes of high-latitude stratospheric SPW1 with equatorial DW2, and finally, panel (d) shows the correlation between equatorial DW2 at 10 hPa with equatorial DW2 at all altitudes. Equatorial DW2 has a very good correlation with high-latitude stratospheric SPW1, particularly near the stratopause and in the MLT region. Simultaneously, the correlation of DW2 in the MLT is also good with that in the lower altitudes over the equator. This indicates that DW2 in MLT is influenced via non-linear interactions by SPW1 from high latitudes as well as other sources from below. In the case of equatorial DS0 in the MLT region, its correlation with high-latitude stratospheric SPW1 does not appear to be significant but is large and positive in the lower mesosphere. The correlation is also reasonably good with itself from the lower altitudes. This shows that both DS0 and DW2 also have a source in the equatorial lower atmosphere and could probably be explained as the global oscillations from classical tidal theory [1–3]. The anti-symmetric nature of their global structures also supports this argument.

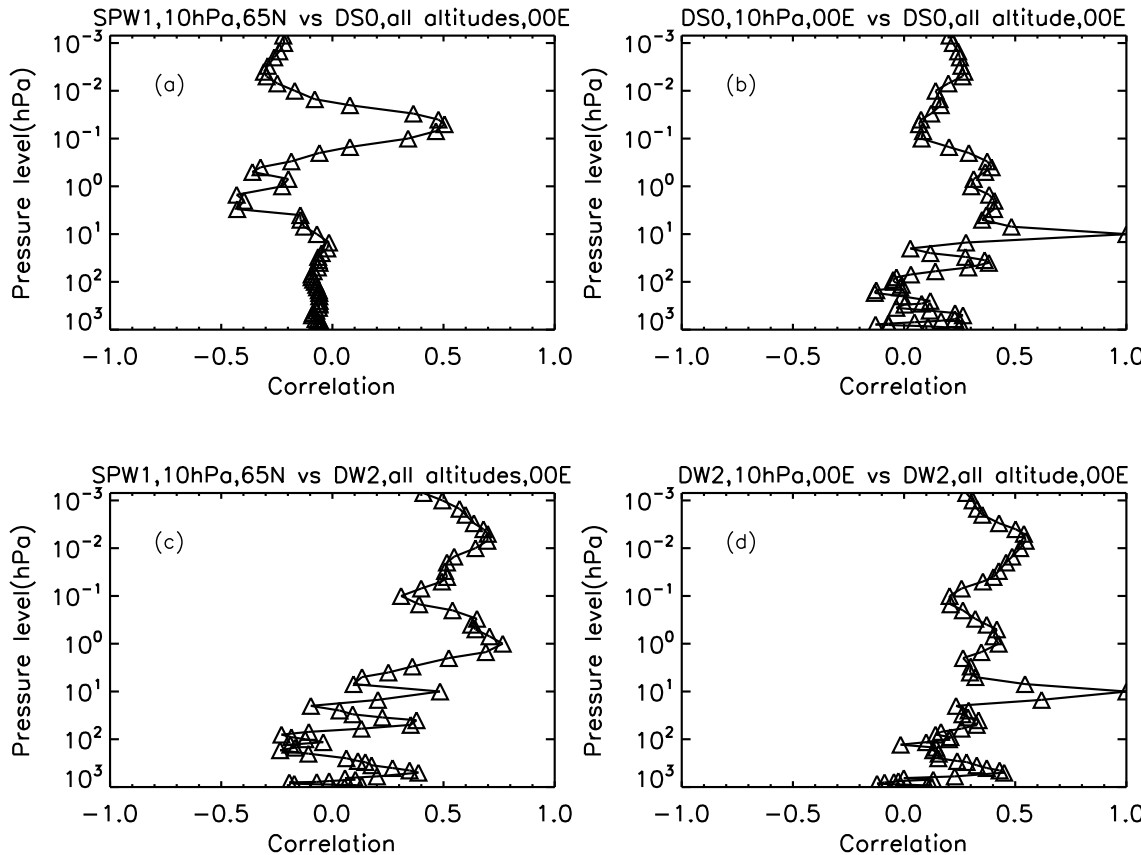

**Figure 11.** Correlation between amplitudes of (**a**) high-latitude stratospheric SPW1 with equatorial DS0, (**b**) equatorial DS0 at 10 hPa with equatorial DS0 at all altitudes, (**c**) high-latitude stratospheric SPW1 with equatorial DW2, (**d**) equatorial DW2 at 10 hPa with equatorial DW2 at all altitudes.

## 5. Summary and Conclusions

Temperature data from FORMOSAT-3/COSMIC satellite observations, the CMAM30 model, and ERA-Interim reanalysis data are analysed to investigate the variabilities in the non-migrating tides DS0 and DW2. Global vertical and horizontal variabilities are analysed, and we conclude the following from the current study.

1.  The tidal variability observed in the stratosphere from CMAM30 and ERA-interim temperature data are highly consistent with each other. Results from COSMIC, however, are comparable in the lower stratosphere over equatorial to mid-latitudes only. This is due to COSMIC data quality issues above 40–45 km and aliasing issues in the high-latitude stratosphere.
2.  Amplitudes of DS0 and DW2 in the high-latitude stratosphere are overestimated in the analysis of satellite datasets due to aliasing and should not be attributed to non-linear interactions. This result is strengthened by the fact that DS0 and DW2 are practically absent in CMAM30 and ERA data in the high latitude stratosphere.
3.  DS0 and DW2 show variabilities ranging from 30 to 100 days over the equator in the MLT region. Part of these variabilities are similar to SPW1 variability during winter in the high-latitude stratosphere of the Northern Hemisphere and support the argument that they could be produced due to non-linear interactions with DW1 in the equatorial MLT region. At other epochs when there is no SPW1 activity, the observed DS0 and DW2 variabilities cannot be explained through this process.
4.  The global structure of DS0 and DW2 tides shows that they have their source in the lower atmosphere, and the tides propagate upwards with increasing amplitudes. The vertical wavelengths of these structures are of the order of $\sim$25 km. The anti-symmetric nature of the vertical global structures indicates that these tides could be the result of global atmospheric oscillations proposed by the classical tidal theory.

**Author Contributions:** Conceptualization, U.D. and S.D.; methodology, S.D.; software, S.D.; validation, S.D. and U.D.; formal analysis, S.D.; investigation, U.D. and S.D.; resources, U.D.; data curation, U.D.; writing—original draft preparation, S.D.; writing—review and editing, U.D.; visualization, S.D. and U.D.; supervision, U.D.; project administration, U.D.; funding acquisition, U.D. All authors have read and agreed to the published version of the manuscript.

**Funding:** This research was funded by Science and Engineering Research Board (SERB), a statutory body of the Department of Science and Technology (DST), Government of India, grant number ECR/2017/002258.

**Institutional Review Board Statement:** Not Applicable.

**Informed Consent Statement:** Not Applicable.

**Data Availability Statement:** All data used in the study are freely avaible for public use. CMAM30 data are dowloaded from https://climate-modelling.canada.ca/climatemodeldata/cmam/output/CMAM/CMAM30-SD/6hr/atmos/ta/index.shtml (accessed on 21 December 2022). ERA-Interim data is downloaded from https://apps.ecmwf.int/datasets/data/interim-full-daily/levtype=sfc/ (accessed on 21 December 2022). The data used in the current study are obtained from UCAR/COSMIC https://cdaac-www.cosmic.ucar.edu/ (accessed on 21 December 2022) (COSMIC Data Analysis and Archive Center, 2013).

**Acknowledgments:** The authors thank Canadian Space Agency for providing free access to CMAM30 and European Centre for Medium-Range Weather Forecasts (ECMWF) for ERA-interim reanalysis data and also acknowledge the UCAR/COSMIC programme for providing free access to FORMOSAT-3/COSMIC 'atmPrf' temperature data. Authors thank all the Reviewers for their valuable comments that have improved the quality of the manuscript.

**Conflicts of Interest:** The authors declare no conflict of interest.

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
