# Peer review of "Short-Term Variability of Non-Migrating Diurnal Tides in the Stratosphere from CMAM30, ERA-Interim, and FORMOSAT-3/COSMIC"

_atmosphere, doi:10.3390/atmos14020265_

Round 1

Reviewer 1 Report

Short term variability of non-migrating diurnal tides in the stratosphere from CMAM30 and ERA-interim, FORMOSAT-3/COSMIC

By Subhajit Debnath1 , Uma Das

Comments:

   The authors use the stratospheric temperature observation data from COSMIC, Model data from CMAM30 and ERA-interim reanalysis data from ECMWF within 2006-2010 to investigate the short-term variabilities of non-migrating tides of DS0 and DW2 with a temporal window of 21d. They obtained that there are significant DS0 and DW2 activities in the equatorial region in COSMIC data which consistent well with CMAM30 and ERA results. There are also significant DS0 and DW2 activities in the winter mid- and high-latitudes in observation data, however they are negligible in CMAM30 and ERA data. They ascribe this difference to the sampling aliasing of the satellite data within 21d window. In addition, they use the mesospheric data from CMAM30 model to investigate the relationship between equatorial non-migrating tides and SPW1 in the high-latitude. They found that during boreal winters, the variability of stratospheric SPW1 over 65°N is similar to that of DS0 and DW2 over the equator at 0.0007 hPa and propose that it is an evidence that SPW1 from high latitude stratosphere moving upward and equatorward could be interacting with the migrating diurnal tide and generating the non-migrating tides in the equatorial mesosphere and lower thermosphere (MLT) as suggested by Lieberman et al (2015). Their researches contribute to the understanding of the short-term variability of non-migrating tides in the stratosphere. The manuscript is clearly presented, and worth to be published. Some minor concerns should be addressed as indicated below before published.

Minor revisions:

(1) On the window size of ±10d , would the authors give an explanation on their choice except for that it is smaller than ~60d for the SABER data analysis? Are there any other reasons? The authors would better add information about the spatial and temporal distribution of COSMIC observation data used in this paper within 21d-window to make sure that the data is suitable for non-migrating tides retrievals.

(2) On the aliasing problem on DS0 and DW2 in observation data only in high latitudes, but not in equatorial regions, would the authors add more explanations?

(3) Page 8 “3.2. Altitude-Latitude Variations” is the same as “Section 3.3. Altitude-Latitude Variations”. I would suggested to delete it.

(4) In Figure 5 and Figure 6, amplitudes of DS0 and TW2 in the equatorial region grow with increasing height both in CMAM30 and ERA data, but they maximize at ~35km in COSMIC observation data. Would the author provide a simple discussion on it? Is it due to the quality of radio occultation data since the statistic optimization is applied in the data retrieval above 30km?

(5) Figure 11: Missing figure number identifiers a-d.

Reviewer 2 Report

Review comments for “Short term variability of non-migrating diurnal tides in the stratosphere from CMAM30 and ERA-interim, FORMOSAT-3/COSMIC” by Debnath et al. submitted to Atmosphere.

Based on satellite observations, model simulations and reanalysis data, this study focuses on the short-term variability of the non-migrating tides DS0 and DW2 in the stratosphere. The results provide a comprehensive picture of the variations of DS0 and DW2 in the stratosphere. Moreover, the results further demonstrate that DS0 and DW2 over the equator in the MLT region are partly due to the nonlinear interactions between SPW1 and DW1, which has been suggested by some previous papers. Overall, the conclusions in this paper are basically supported by the results. Therefore, I recommend this paper be published after some revisions.

My main concern with this study is the use of FORMOSAT-3/COSMIC observations. In Section 5, the authors suggest that the tidal variability extracted from COSMIC, CMAM30, and ERA-interim is consistent, but I disagree. Based on the results in Figures 1-6, I can only find that the tidal variability in CMAM30 and ERA-interim is similar, and the results from COSMIC have large discrepancies from the other two data sets. As shown in Figures 5 and 6, the spatial structures of DS0 and DW2 in COSMIC observations and the other two data sets are completely different since the peaks of the tidal amplitudes are located at different latitudes/altitudes, which makes the comparisons between them meaningless. The results in Figure 7 over the equator are the only comparable ones in this paper. Moreover, the discussion part is totally based on the results from CMAM30 shown in Figures 8-10. Therefore, I think that the results from FORMOSAT-3/COSMIC observations contribute little to the main conclusion of the article, and the extraction of tides from satellite observations is inherently subject to large errors. The authors should consider deleting the results from COSMIC, or reorganize the results so that they are somewhat comparable to the model simulations and reanalysis data.

In addition, the language and grammar of this paper also need further revision.

Reviewer 3 Report

Review of “Short term variability of non-migrating diurnal tides in the stratosphere from CMAM30 and ERA-interim, FORMOSAT-3/COSMIC” by Debnath and Das.

The manuscript deals with the variability observed in non-migrating tidal components DS0 and DW2 observed using a set of different techniques. The results shed some light on the issue of aliasing involved in the tidal analysis satellite observations.  However, to a major part, the results are presented in a vague manner and it is often not clear to the reader what the authors want to convey with the analysis. Therefore I recommend the manuscript to undergo a major revision before it can be considered for publication. The following are my concerns:

Major Comments:

1.       What is the reason behind choosing individual days for discussing the latitude-height structure of DS0 and DW2 tides in Figures 5 and 6? Tides have large day-to-day variations. As the authors themselves state, tides undergo strong interaction with planetary waves and other tidal components, which is another major cause of their day-to-day variability. In such a case, do the authors think similar patterns would arise for other days? Why were these days and years chosen? If I am not wrong, 36th day of 2009 is very close to a major Sudden Stratospheric Warming during which planetary wave activity in the high latitudes(and also tidal activity the low latitudes) is much more dramatic. Do the authors want to comment on this? In any case, the authors should justify the choice of their dates or demonstrate that they are representative of the respective seasons by showing the seasonal or monthly averages.

2.       What are the errors associated with these observations? More often than not, tidal amplitudes shown in the manuscript are of the order of 1K. Are they statistically significant?

3.       Why is the amplitude and phase structure of DS0 drastically different in CMAM 30 and ERA interim results? Even the vertical wavelength seems to be different. Are the same tides even observed here?

4.       Is there a gap in CMAM temperature data in 2010 (probably May-June)? See Figures 1 and 2 for the vertical blank spaces. This probably results in artefacts in Figure 8 for DS0 at 0.0007 hPa. If it is a gap in temperature data, why does it not appear in DW2 amplitudes and phases?

5.       “Das [37] showed that SPW1 amplitudes in SABER and COSMIC temperatures show variability of 30 to 120 days. The smaller period variabilities are seen in COSMIC using a window size of 21 days and thus clearly indicate that it can impact the accurate deduction of DS0 and DW2 variabilities. It is thereby concluded that there is high probability that the high DS0 and DW2 amplitudes observed over high latitudes in COSMIC could be due to aliasing.” What do the authors mean by this? Do the authors mean that the presence of smaller period variabilities imply the presence of aliasing?

Apart from this, if aliasing is a problem only in high latitudes, then why is the altitude structure of DS0 and DW2 tidal amplitudes dramatically different in COSMIC and CMAM/ERA interim results?

6.       Can you plot amplitudes instead of wavelet power in Figure 9? With wavelet power, it becomes difficult to conceptualize what is happening.

7.        What is the rationale behind choosing 10hPa for estimating the correlations in tides in Figure 11? Do you expect to get any physical insights into the behaviour of tides by correlating tides at 10hPa with themselves at other altitudes? Moreover, what data has gone into estimating the correlations in Figure 11? Is it the whole time series?

8.       I don’t see anything meaningful coming out of the wavelet analysis. What I see in the discussion merely a reiteration of what is seen in the figure. Vague statements like ‘sometimes similar periods exist and sometimes do not’ should be avoided.

Minor Comments

1.       Please include line numbers in all future versions of the manuscript. Without line numbers, the review process becomes painful.

2.       The authors need to clearly describe what they mean by aliasing in the manuscript text. Otherwise, readers outside of the tidal community will be clueless.  

3.       What follows the question “How are these non-migrating tides generated over the equatorial and low latitude regions?” in Page 16 confuses the readers. Rephrasing is needed.

4.       Why CMAM30 is only available from 1979 to 2010? Is it not a model which can be run for any period of time? I guess it is that the model runs you used are available only for this period.

5.       “The amplitudes are comparatively smaller in 2007. This could be due to a modulation in the equatorial QBO.” Why specifically QBO? If it the effect of QBO, it should be visible in other years also. Justify your statement with appropriate reasoning or references.

6.       Why is the colorbar in many of the plots restricted to 1K, while the magnitudes are still greater than 1K?

7.       Remove “3.2 Altitude-latitude variations in Page 8”

8.       Plot titles in Figure 7 says “for 21 days”. Is it true? Or do the authors mean a 21-day window?

9.       Title may be reworded for betterment. “Short term variability of non-migrating diurnal tides in the stratosphere from CMAM30, ERA-interim and FORMOSAT-3/COSMIC”

10.   Without presenting any analysis, the authors argue that the tidal structures arise from global atmospheric oscillations predicted by the tidal theory. Please explain what the authors mean by global atmospheric oscillations.

Round 2

Reviewer 3 Report

All my concerns have been addressed in the revised manuscript. I recommend accepting the manuscript in present form.